# Bacterial Type I Toxins: Folding and Membrane Interactions

**DOI:** 10.3390/toxins13070490

**Published:** 2021-07-14

**Authors:** Sylvie Nonin-Lecomte, Laurence Fermon, Brice Felden, Marie-Laure Pinel-Marie

**Affiliations:** 1CiTCoM, CNRS, UMR 8038, Université de Paris, 93526 Paris, France; sylvie.nonin@u-paris.fr; 2BRM (Bacterial Regulatory RNAs and Medicine), Inserm, UMR_S 1230, Université de Rennes 1, 35000 Rennes, France; laurence.fermon@univ-rennes1.fr (L.F.); brice.felden@univ-rennes1.fr (B.F.)

**Keywords:** toxin-antitoxin systems, type I toxins, mechanisms of action, membrane depolarization, membrane permeabilization, pore formation, nucleoid condensation, structure, folding

## Abstract

Bacterial type I toxin-antitoxin systems are two-component genetic modules that encode a stable toxic protein whose ectopic overexpression can lead to growth arrest or cell death, and an unstable RNA antitoxin that inhibits toxin translation during growth. These systems are widely spread among bacterial species. Type I antitoxins are *cis*- or *trans*-encoded antisense small RNAs that interact with toxin-encoding mRNAs by pairing, thereby inhibiting toxin mRNA translation and/or inducing its degradation. Under environmental stress conditions, the up-regulation of the toxin and/or the antitoxin degradation by specific RNases promote toxin translation. Most type I toxins are small hydrophobic peptides with a predicted α-helical transmembrane domain that induces membrane depolarization and/or permeabilization followed by a decrease of intracellular ATP, leading to plasmid maintenance, growth adaptation to environmental stresses, or persister cell formation. In this review, we describe the current state of the art on the folding and the membrane interactions of these membrane-associated type I toxins from either Gram-negative or Gram-positive bacteria and establish a chronology of their toxic effects on the bacterial cell. This review also includes novel structural results obtained by NMR concerning the *sprG1*-encoded membrane peptides that belong to the *sprG1*/SprF1 type I TA system expressed in *Staphylococcus aureus* and discusses the putative membrane interactions allowing the lysis of competing bacteria and host cells.

## 1. Introduction

Bacterial toxin-antitoxin (TA) systems are two-component genetic modules that encode a stable toxic protein, whose ectopic overexpression can lead to growth arrest or cell death, and an unstable antitoxin, which neutralizes toxin activity during bacterial growth [1]. TA systems are widely spread among bacterial genomes, highlighting their potential importance [2]. They are classified into seven types depending on the antitoxin nature and its mode of action. While the toxins are always proteins, the antitoxins can be either non-coding RNAs (in type I and III systems) or small proteins (in types II, IV, V, VI, and VII). Antitoxins act by inhibiting toxin synthesis (in types I and V), sequestering the toxin (in types II and III), counteracting toxic activity (in type IV), promoting toxin degradation (in type VI), or by chemical modification of the toxin at a post-translational level (type VII) [3]. Recently, a novel type of TA system has been proposed in which both the toxin and the antitoxin are small RNAs [4]. Moreover, a type II/IV TA hybrid system has been described in which the DarG antitoxin interacts with the DarT toxin to inhibit its expression, but also acts on the target of DarT [5,6]. Antitoxins are unstable and more susceptible to degradation by ribonucleases or proteases than toxins, leaving toxins free to interfere with essential cellular functions such as replication, translation, or cell division [7]. In this review, we will focus exclusively on type I TA systems, characterized by a small antisense RNA that base-pairs with its cognate toxin-encoding messenger RNA (mRNA) to prevent the toxin synthesis under normal growth conditions. These systems are first predicted in the bacterial genomes through computational approaches [8]. However, only a few of them are experimentally characterized and mostly focused on *E. coli*.

Type I TA systems were initially identified on plasmids, where they ensure plasmid maintenance through a post-segregational killing (PSK) mechanism [9]. In this process, because of the antitoxin degradation, the plasmid loss results in the decrease of the antitoxin cell concentration and the killing of the plasmid-free cell by the stable toxin. Later on, homologues of known type I TA systems were discovered on bacterial chromosomes [8,10]. Although the biological role of chromosomal type I TA systems is still elusive, they can be involved in mobile genetic elements maintenance, growth adaptation to environmental stresses, or persister cells formation [11]. Persister cells represent a subpopulation of genetically identical and metabolically slow-growing bacteria that are tolerant to extremely high antibiotic doses after selection through repeated antibiotic therapy. They can result in treatment failure, relapse, and persistent bacterial infections [12].

Type I toxin expression is tightly controlled by RNA antitoxins and *cis*-encoded mRNA functional elements [13]. Most type I antitoxins are *cis-*acting antitoxins (e.g., *hok*/Sok, *bsrG*/SR4, *sprG*/SprF), meaning that the antitoxin and toxin loci overlap, resulting in a perfect pairing between the two RNAs. The *trans*-acting antitoxins (e.g., *tisB*/IstR1, *dinQ*/AgrB) are located away from the toxin locus and share often limited sequence complementarities. The detailed description of the influence of the toxin expression on the RNA antitoxin regulation is not within the scope of our analysis. It has already been covered by two excellent reviews which describe type I antitoxin mode of action [13,14]. Toxin-antitoxin RNA duplex formation can either result in toxin mRNA degradation or, more commonly, in toxin translation inhibition, or in the combination of these two regulatory mechanisms [14]. In the cases where the antitoxin binding is not efficient enough to abolish toxicity, *cis*-encoded mRNA elements sequestering the ribosome binding site (RBS) are also required for toxin repression [13].

We recently classified the type I toxins into two categories: membrane-associated type I toxins and cytosolic type I toxins [15]. The membrane-associated type I toxins generally contain less than 60 amino acids, are hydrophobic, and have a putative α-helical transmembrane domain. For many of these type I toxins, the toxic activity is linked to membrane depolarization and/or permeabilization, followed by an intracellular ATP depletion [15]. The cytosolic type I toxins are RalR and SymE toxins that promote DNA or RNA cleavage, respectively [16,17].

In this review, we will focus on the membrane-associated type I toxins from either Gram-negative or Gram-positive species with a characterized mechanism of action. We describe the current state of the art on protein folding and membrane interactions of these bacterial toxins and we will attempt to establish a chronology of their toxic effects on the bacterial cell. As a new result, we will also investigate, by NMR, the structure of the *sprG1*-encoded membrane peptides that belongs to the *sprG1*/SprF1 type I TA system expressed in *S. aureus* and discuss the putative membrane interactions responsible to the lysis of competing bacteria and host cells.

## 2. Overview of the Membrane-Associated Type I Toxins across the Bacterial Species

The membrane-associated type I toxins have only been described in Proteobacteria and Firmicutes [14]. We have classified the membrane-associated type I toxins whose structure and/or mechanism of action have been deciphered (Table 1). These small hydrophobic peptides like phage holins [8] or cationic antimicrobial peptides [18] all display an α-helical transmembrane domain but have a strong diversity in length and amino acid sequence. The α-helical structure was experimentally validated for some of them such as TisB, IbsC, and LdrD in *Escherichia coli* [19,20,21], Fst in *Enterococcus faecalis* [22], SprA1 and SprG1 in *S. aureus* ([23] and in this study), and AapA1 in *Helicobacter pylori* [24] (Figure 1). For the other type I toxins, we could predict the α-helical transmembrane domain using the TMPRED algorithm along with the orientation of the computed in silico model of α-helix from the inside (I) to the outside (O) of the bacterial membrane with a given probability ranging from ++ to − in decreasing order (https://embnet.vital-it.ch/software/TMPRED_form.html, accessed on 5 April 2021) (Table 1). Among the common features, the C-terminal part is mainly predicted to be localized in the cytosol, except for the ShoB, HokB, SprA1, and SprG1 toxins, and has positively charged residues, except for IbsC, ShoB, and DinQ (Table 1). These positively charged residues were shown to be crucial for the binding/anchoring of peptides on lipid bilayers and may help interactions with the bacterial membrane that is mostly negatively charged [25]. The presence of cysteine residues in HokB, ShoB, AapA1, and SprA1 suggests a possible dimerization/oligomerization of these toxins, leading to pore formation. Sequence requirements for toxicity have been demonstrated by mutagenesis approaches for the Fst, IbsC, and AapA1 toxins [20,24,26]. These studies revealed that only a few residues are critical for toxicity, but the lack of sequence conservation within the type I toxins impedes the prediction of amino acids essential for toxicity by homology searches using bioinformatics tools [8]. Despite several common features, the membrane-associated type I toxins exert distinct mechanisms of action based on toxin ectopic overexpression. We can distinguish type I toxins whose overexpression induces bacterial membrane alterations as primary detected and surely toxic effects such as membrane depolarization and/or permeabilization, and the type I toxins whose overexpression induces morphological changes in bacteria as a primary detected effect prior to membrane perturbations.

## 3. Membrane-Associated Type I Toxins Inducing Membrane Perturbations as a Primary Detected Effect

### 3.1. Membrane-Associated Type I Toxins Inducing Pore Formation

The two well-characterized membrane-associated type I toxins HokB and TisB disrupt membrane integrity though pore formation in *E. coli*. The ectopic overexpression of these two pore-forming toxins causes membrane depolarization and generates dead cells so-called “ghost cells” with an unusual morphology characterized by cell material at the poles and a translucent cell center [9,30].

The Hok type I toxin is expressed from the *hok/Sok* locus that is the first type I TA system discovered on the R1 plasmid in *E. coli* where it confers plasmid maintenance through PSK [9]. The *hok/Sok* locus codes for three genes: *hok* (for host killing), *sok* (for suppression of killing), and *mok* (for mediation of killing). The *hok* gene encodes a 52 amino acid membrane-associated peptide whose ectopic overexpression provokes loss of membrane potential and arrest of respiration, and kills bacteria within 30 min upon induction (Figure 2) [9]. The activation of *hok* translation in R1 plasmid-free cells is linked not only to the absence of the unstable Sok RNA antitoxin but also to the 3’-end processing and refolding of the *hok* mRNA, which increase its activity and stability [31]. In plasmid-carrying cells, the Sok RNA antitoxin binds to the *mok-hok* mRNA at the level of the Shine–Dalgarno (SD) sequence of *mok* encoding the Mok peptide, upstream of *hok*. The degradation of this RNA duplex by the RNAse III prevents translation of the *hok* mRNA, leading to cell growth [31]. Five *hok/Sok* homologues were discovered later in the *E. coli* chromosome, but many of them are non-functional due to mutations, insertions, or large rearrangements [10]. By single-cell approach, the Michiels group demonstrated that the GTPase ObgE promotes *E. coli* persistence through transcriptional activation of *hokB* expression, requiring the stringent response alarmone (p)ppGpp [32]. The GDP or ppGpp binding seems to be required for the GTPase ObgE to induce *hokB* transcription [33]. The deletion of *hokB* does not impact *E. coli* persistence, but the ectopic overexpression of *hokB* increases the number of persister cells in response to ofloxacin or tobramycin exposure [32]. The relationship between the mode of action of the 49 amino acid membrane-associated peptide HokB and *E. coli* persistence was elucidated in 2018 [34]. Using in vitro conductance measurements with synthetic or natural planar lipid bilayers, the authors showed that HokB targets the lipid bilayer and forms pores with an estimated diameter of 0.59–0.64 nm [34]. Changes in conductance using specific PEGylation demonstrated that HokB crosses the lipid bilayer with its positively charged N-terminal domain extending in the cytoplasm, whereas the negatively charged C-terminal domain extends in the periplasm [34]. Thanks to a m-Cherry tag at the N-terminal domain, the authors showed by microscopy cluster formation of HokB peptides at the *E. coli* membrane, consequently to an in vivo pore formation. By blocking these HokB pores with PEG 1000 in vivo, the group elegantly proved that pore formation by HokB is directly linked to *E. coli* persistence and results in a decrease of the cellular energy ratio and the ATP efflux within 4 h upon induction. Moreover, the authors demonstrated in vitro that the membrane potential controls the size of HokB pores since, when the potential applied to the membrane is high, mature pores are formed and provoke both a membrane depolarization and an ATP efflux, leading to persistence in metabolically active cells (Figure 2). When the applied potential is low, intermediate pores can be formed. They induce ATP/ADP ratio drop, but no persister cells formation [34]. HokB overexpression triggers membrane depolarization and reactive oxygen species (ROS) formation within 1 h upon induction, leading to growth inhibition which could contribute to persister cells formation [35] (Figure 2). Recently, mechanisms that control the formation and the awakening of HokB-induced persister cells have been identified [36]. HokB contains three cysteine residues of which cysteines C9 and C14 are predicted to be part of a transmembrane α-helix anchored to the membrane and cysteine C46 to be present in the periplasmic C-terminal domain (Table 1). Construction of cysteine-to-serine substitution mutants demonstrated that the periplasmic cysteine C46 is responsible for HokB dimerization by disulfide bridge formation as well as for HokB-induced membrane depolarization and persister cells formation [36]. The inter-peptide disulfide bridge formation between two periplasmic C46 residues is mediated by the periplasmic oxidoreductase DsbA. This enzyme is essential for HokB dimerization and stability and, consequently, for pore formation, membrane depolarization, ATP efflux, and persister cells formation [36]. A positive correlation between the concentration of HokB peptides and the dormancy duration in persister cells was evidenced by microfluidics and single-cell approaches. It suggests that pore disassembly may be involved in the awakening of HokB-induced persister cells. Upon awakening, the periplasmic oxidoreductase DsbC was shown to reduce the disulfide bridges and induce HokB monomerization, thus promoting DegQ-mediated degradation of HokB peptides. This pore destabilization leads to membrane repolarization and ATP production by the NADH-dehydrogenase complex I and to resume the growth of HokB-induced persister cells [36].

The TisB type I toxin is another pore-forming peptide expressed in *E. coli* from the *tisB/IstR-1* locus that was discovered by the first genome-wide searches for bacterial small RNAs [37,38]. The Wagner group demonstrated that *tisB*/IstR-1 is the first type I TA system involved in the SOS response [39]. The SOS response is initiated when the recombinase RecA senses DNA damage and activates cleavage of the global repressor LexA [40]. The *tisB* (for toxicity induced by SOS) gene encodes a 29-amino-acid peptide located in the inner membrane of *E. coli* [30]. Its ectopic overexpression provokes membrane damages and a rapid drop in ATP levels that results to a drastic decrease in transcription, translation, and replication rates, leading to cell death within 1 h upon induction (Figure 2) [30,39]. Under normal growth conditions, the transcription of *tisB* is repressed by LexA. When SOS conditions are encountered, RecA-induced cleavage of LexA derepresses *tisB* [39]. The high *tisB* mRNA levels out-titrate the IstR-1 (for inhibitor of SOS-induced toxicity by RNA) RNA antitoxin pool, despite its constitutive expression, and promotes cell growth arrest [39]. It was proposed that the IstR-1 antitoxin prevents inadvertent toxicity that results from leaky *tisB* transcription. IstR-1 RNA base-pairs with the standby site of the active +42 *tisB* mRNA and promotes RNA duplex degradation by RNase III to an inactive +106 *tisB* mRNA, thus preventing ribosome access and toxin translation [41]. The effects of TisB on the membrane integrity, decreasing intracellular ATP levels and all essential cellular processes [30], were associated to *E.coli* persistence in 2010. The Lewis group showed that ciprofloxacin, a DNA-damaging antibiotic targeting DNA gyrase and topoisomerase IV, increases persister levels via SOS-dependent induction of TisB [42]. A strain deleted for the *tisB* locus showed a 10-to-100-fold decrease of ciprofloxacin-tolerant persisters, whereas deletion of the *istR-1* locus resulted in a 10-to-100-fold increase [42]. Moreover, ectopic overexpression of TisB confers multidrug tolerance. The authors hypothesized that, in the presence of DNA-damaging antibiotics, the optimal strategy to survive is dual encompassing DNA repair activation and switching to a dormant state via the SOS-induced TisB expression [42]. The Wagner group demonstrated that the deletion of two regulatory RNA elements (i.e*.,* Ist-R1…) and the inhibitory 5’ UTR structure in the *tisB* mRNA [41] triggers stochastic TisB translation, leading to pore formations and superoxide production [35,43]. This results in a depolarized sub-population that turns into persister cells even in the absence of strong SOS induction by exposure to ciprofloxacin [43,44]. The deletion of the superoxide dismutases SodA and SodB impairs the stochastic TisB-dependent persisters’ formation and the recovery of persister cells after ciprofloxacin exposure [35]. In 2012, two studies investigated the structure of TisB and its interaction with model of bacterial membranes. The first study showed by in vitro conductance measurements and polymer-exclusion experiments that TisB forms narrow anion-selective pores in planar lipids bilayers as consequence of the net-positive charge of TisB [45]. The other study confirmed the amphiphilic α-helical conformation of TisB by circular dichroism, oriented circular dichroism, and molecular dynamic (MD) simulations. The authors showed a spontaneous insertion of TisB in lipid bilayers with a stable transmembrane alignment where the four charged side-chains of D5, K12, D22, and K26 lie on a narrow strip along with the polar face of the helix, with an alternating pattern of positive and negative charges. They also postulated formation of a spontaneous assembly as antiparallel dimers stabilized by four intermolecular salt bridges and an intermolecular hydrogen-bond interaction between Q19 residues (Table 1) [19]. The polar interface of the TisB helix could mediate the passage of protons across the hydrophilic lipid bilayer. Fluorescence dequenching experiments confirmed that TisB can form narrow pores of an estimated diameter of 0.15 nm that are impermeable to intracellular water-soluble components, thus allowing cell survival [19,45]. The flows of protons and anions across TisB pores can dissipate the proton motive force (PMF), reduce the ATP production, and hijack the metabolism to a dormant state (Figure 2). Consequently, TisB leads to the shutdown of the major antibiotic targets and induces multidrug tolerance.

### 3.2. Membrane-Associated Type I Toxins Inducing Membrane Depolarization and/or Permeabilization

For some membrane-associated type I toxins, the ectopic overexpression rapidly induces membrane depolarization and/or permeabilization, but the mechanism of action has not been clearly demonstrated by biophysical approaches.

This is the case for the SprA1 (for Small pathogenicity island RNA) type I toxin (also named PepA1) of the *sprA1*/SprA1_AS_ TA system expressed in *S. aureus* and that displays homology with the Fst/Ldr family [8,46,47]. SprA1 is a 30-amino acid peptide located in the membrane whose ectopic overexpression causes *S. aureus* cell death through membrane permeabilization within 1 h upon induction (Figure 2) [23]. The *sprA1* mRNA expression is prevented in *trans* by the SprA1_AS_ *cis*-RNA antitoxin that base-pairs with the internal RBS and thus inhibits the toxin translation [46]. Scanning electron microscopy (SEM) shows that synthetic SprA1 provokes the lysis of competing bacteria, such as *S. aureus* and *E. coli,* and of human erythrocytes [48]. Consequently, in contrast to Fst and AapA1 toxins [22,24], SprA1 has an antibacterial action when added to the extracellular medium, supporting the idea that its primary toxic effect flows from its interaction with the membrane. The 3D model of the structure of SprA1 performed by NMR supports this hypothesis. SprA1 folds into an extended amphipathic transmembrane α-helix that is interrupted at the P10 proline residue and slightly bends at the C15 cysteine residue in the conserved PXXXGC motif of the Fst/Ldr family [23,49]. The N-terminus of SprA1 is unfolded and the positively charged C-terminus is folded (Table 1, Figure 1). In silico MD simulations show that, when inserted in a DPPC bilayer solvated by water molecules, the NMR SprA1 structure rapidly changes its conformation into an uninterrupted extended α-helix (Figure 1) [23]. SprA1 also releases the cytoplasmic content of cells like the cationic antimicrobial peptide nisin, known to form pores leading to *S. aureus* cell death. The authors hypothesized that SprA1 could be a pore-forming toxin [48]. Moreover, the presence of the C15 at a flexible hinge in the α-helix suggests a possible dimerization that could help to form pores as shown for HokB [34]. Future biophysical studies are needed to decipher the mechanisms of action responsible for the permeabilization of both prokaryotic and eukaryotic cell membranes. The expression of SprA1 is induced upon acidic or oxidative stress in response to a drastic decrease of SprA1_AS_ levels [23]. One hypothesis is that, after its internalization by phagocytes that triggers deleterious oxidative and acidic bursts, *S. aureus* would induce SprA1 expression to disrupt the phagolysosome and cell membranes to escape the immune cells, kill competing bacteria, and spread into the host.

Another example is the Lpt toxin, a 29-amino-acid peptide expressed from the type I TA system RNAI/RNAII identified in the plasmid DNA of *Lactobacillus rhamnosus* by its sequence homology with Fst (Table 1) [50]. Induction of Lpt expression in *E. coli* causes growth arrest, nucleoid condensation, and membrane permeabilization within 1 h upon induction (Figure 2). However, 3 h after induction, the *E. coli* growth resumes concomitantly with the recovery of nucleoid compaction and membrane damages, highlighting that the toxin overexpression leads to cell stasis instead of cell death [51]. Both morphological modifications and membrane perturbations effects have been observed at least 1 h after induction. However, surface visualization by AFM of Lpt-expressing *E. coli* cells evidence membrane patches linked to a disruption of the lipid bilayer. The authors suggest that Lpt acts through a detergent-like mechanism where the peptide, when added at high concentration, micellizes the lipid bilayer [51]. Despite these observations, it is not excluded that cell morphology changes induced by Lpt overexpression could be the primary triggers of its toxic effect. The membrane localization of Lpt was evidenced by the fusion of the red fluorescent protein m-Cherry to the C-terminus of the peptide. This fusion or the removal of the hydrophilic C-terminal region abolishes the toxicity of Lpt, supporting the importance of the C-terminal domain for toxicity. Moreover, the substitution of a proline (P11) by a charged glutamic acid in the α-helical transmembrane domain also suppresses the toxicity of Lpt, suggesting that the distortion of the α-helix induced by P11 is essential for the interaction with the membrane and thus, for the toxicity, as this is the case for Fst [26]. Although Lpt, Fst, and SprA1 all share the conserved motif APXXXGXXX, where X represents hydrophobic amino acids (Table 1) [23,52], it seems difficult to predict a similar mechanism of action for these three toxins. Indeed, the SprA1 homologue, SprA2 (also named PepA2), that displayed the conserved PXXXGC motif, does not lyse bacterial cells extracellularly but has a more hemolytic effect than SprA1 [13]. Lpt induces cell morphology changes and it is not excluded that this could be the primary trigger of its toxic effect. For this TA system, the molecular mechanisms involved in the regulation of Lpt expression by the RNA II antitoxin are currently unknown.

Five other type I toxins expressed by *E. coli* (namely ZorO, IbsC, ShoB, LdrA, and DinQ) rapidly cause membrane depolarization and/or permeabilization and/or intracellular ATP drop when overexpressed. The ZorO type I toxin was identified by bioinformatics with search parameters including tandemly duplicated, small, hydrophobic proteins by the Fozo group [8]. ZorO (for Z-protein often repeated) is a 29-amino-acid peptide (Table 1) whose expression is tightly repressed by a 5’-UTR secondary structure that sequesters the RBS of *zorO* mRNA and by the OrzO (for Overexpression reduces Z protein toxicity) RNA antitoxin that base pairs to the EAP (for Exposed After Processing) region located upstream of the RBS of *zorO* mRNA [11,53]. When overexpressed in *E. coli*, ZorO localizes at the inner membrane and induces membrane depolarization associated with a reduction of the intracellular ATP level and an increase of ROS production, leading to the membrane permeabilization and finally to cell death within 30 min (Figure 2) [8,54]. A mutagenesis analysis on ZorO demonstrated that only the five amino acids “ALLRL” spanning positions 20 to 24 are necessary for toxicity while the 13 N-terminal amino acids are dispensable [54]. Conversely to full-length ZorO, the ALLRL peptide, when added to the extracellular medium, exhibits antimicrobial activity against the Gram-positive bacteria *S. aureus* and *B. subtilis* and against the fungus *C. albicans* through a mechanism of membrane permeabilization. However, it does not inhibit *E. coli* growth, suggesting that the outer membrane of Gram-negative bacteria prevents its binding to the inner membrane [54]. These observations indicate that accessibility to the inner membrane is essential for ZorO to exert its toxicity against *E. coli*.

The IbsC (for Induction brings stasis) and ShoB (for Short hydrophobic ORF) type I toxins from the *ibsC*/SibC and *shoB*/OhsC TA systems were discovered in *E. coli* by a bioinformatics analysis designed to identify new TA systems [55]. IbsC and ShoB are, respectively, 19- and 26-amino-acid peptides whose expression is prevented by SibC (for Short intergenic abundant sequence) and OhsC (for Oppression of hydrophobic ORF by sRNA) RNA antitoxins [56]. SibC interacts with the *ibsC* mRNA through the TRD1 (for Target Recognition Domain) and TRD2 domains to repress IbsC toxicity [57]. IbsC and ShoB overexpressions induce membrane depolarization within 5 min upon induction followed by 20 min of incubation with dye (50% and 98% of depolarized cells for IbsC and ShoB, respectively), leading to a massive decrease of cell viability within 30 min upon induction (Figure 2) [56]. IbsC and ShoB are predicted to display a transmembrane domain. The formation of an α-helix has been confirmed by CD experiment for IbsC (Table 1) [20]. A mutagenesis approach revealed that the IbsC sequence can be reduced to 15 amino acids with a minimum of 10 hydrophobic residues to retain toxicity and probably keep the helix translocation across the inner membrane. This minimal sequence requires the C-terminal residues and the hydrophobic residues near the center of IbsC to keep toxicity (residues 6 to 19), suggesting that these residues are probably involved in the promotion of a proper transmembrane 3D conformation and/or protein–protein interactions. On the contrary, the N-terminal residues deletion did not impact the toxicity of IbsC, except for V5 [20]. Whole genome expression analyses carried 20 min after IbsC, ShoB, LdrD, or TisB overexpressions showed the induction of a common set of genes encoding membrane proteins and/or proteins involved in sugar transport and in stress response, as the *soxS* mRNA encoding a regulator of the superoxide stress response [56]. These observations are not surprising as these type I toxins are localized on membrane and induce cell toxicity. The individual overexpression of each toxin also induces the expression of specific genes. Interestingly, IbsC overexpression specifically increases the expression of the *pspABCDE* operon where PspA protein, known to be induced by pore-forming proteins [58], can interact with phospholipids to block the passage of protons across the damaged membrane [56]. Moreover, ShoB overexpression up-regulates the expression of genes involved in sugar transport. These gene regulations could be linked to a direct effect of the toxins on membrane or metabolism or to a secondary consequence of membrane depolarization [56,59].

The 35-amino acid LdrA toxin, which belongs to the Ldr (for Long direct repeat) family of type I TA systems, rapidly induces intracellular ATP drop within 2 min upon induction. This results in a simultaneous inhibition of DNA replication, transcription, and translation within 10 min upon induction and, finally, to cell growth arrest within 30 min upon induction (Figure 2) [60]. This timing strongly suggests a direct effect of LdrA on the integrity of the membrane. LdrA forms an α-helix confirmed by circular dichroism and is predicted by MD simulations to cross the inner membrane of *E. coli* with positively charged C-terminal domain located in the cytosol (Table 1) [60]. The toxicity of LdrA is abolished when a His-tag is added at the N-terminal domain, suggesting the importance of this domain for the anchoring in the inner membrane, leading to the inhibition of ATP synthesis [60]. It can be noted that LdrA is the one of few type I toxins for which the extracellular N-terminal part has been demonstrated to be essential for its toxicity. As for Lpt, the molecular mechanisms involved in the regulation of LdrA expression by its antitoxin remain to be deciphered.

The DinQ type I toxin from the *dinQ*/AgrB TA system is a 27-amino-acid peptide identified in *E. coli* as a new gene of the LexA regulon involved in the SOS response to DNA damaging agents like UV [37,57,61]. DinQ is located within the inner membrane of *E. coli* and its ectopic overexpression induces membrane depolarization within 5 min upon induction followed by 20 min of incubation with the dye, leading to cell death and a high UV sensitivity counteracted by the AgrB RNA antitoxin [62]. AgrB base pairs to the *dinQ* translationally active +44 transcript, resulting in the sequestration of the SD sequence, the cleavage of the duplex by RNase III, and, consequently, preventing ribosome access and toxin translation [63]. These observations have been supported in the *agrB* mutant, whose DinQ overexpression displays not only an intracellular ATP drop before and after UV exposure, an increase of UV susceptibility, but also an impairment of conjugal recombination and an increase of cells with a compacted nucleoid [62]. The authors suggest that DinQ affects the transformation of the nucleoid morphology in response to UV damage and that it could regulate at the inner membrane the DNA repair of hyperstructures associated with homologous recombination, as an additional effect of membrane perturbation [62]. As demonstrated for HokB, TisB, and ZorO, DinQ overexpression also provokes ROS formation within 1 h upon induction, which seems to be a key event for these toxins promoting cell death and which may be linked to pore formation (Figure 2) [35]. Structure predictions revealed that DinQ can form a transmembrane α-helix where the two positively charged lysine residues (K4 and K9) are close to the phospholipid head groups and where E17, R20, and Q24 residues may form a polar patch that can interact with other membrane proteins (Table 1) [62]. As a conclusion, the primary toxic effect of DinQ is controversial: some clues suggest that DinQ is a membrane-disrupting toxin [58], while others suggest that DinQ is involved in nucleoid compaction [64].

Finally, we described here eight toxins whose overexpression leads to a membrane perturbation as a primary detected effect. For two of them, the cause of this membrane perturbation is well investigated and due to pore formation. We also presented two other toxins, Lpt and DinQ, which are suspected to target the membrane as a first toxic effect but whose overexpression also leads to morphological changes detected at the same time. More information about the timing of the effect of toxins overexpression is needed to conclude. To note, most of these toxins are found in *E. coli* genomes, which could be a bias because *E. coli* has been more studied than other bacteria.

## 4. Membrane-Associated Type I Toxins Inducing Cell Morphology Changes as a Primary Detected Effect

Some membrane-associated type I toxins affect bacterial inner membranes, but cell morphology changes are induced as primary detected effects prior to membrane damages. This class of type I toxins is represented by Fst, BsrG, AapA1, and LdrD toxins and probably by Lpt and/or DinQ toxins as discussed before.

The Fst (for faecalis plasmid stabilizing toxin) type I toxin is a 33-amino-acid peptide that belongs to the RNAI/RNAII PSK TA system, expressed from the *par* locus of the *Enterococcus faecalis* plasmid pAD1 [65]. The Weaver group demonstrated that, in *E. faecalis* and in the presence of the pAD1 plasmid, the interaction between RNAI, the messenger RNA encoding Fst, and the antitoxin RNAII is initiated at the loop of the intrinsic terminators and that it spreads to the DRa (for DNA direct repeats) and DRb sequences located at the 5’ end of both RNAs. This results in a sequestration of the GUG initiation codon of RNAI and subsequently to the inhibition of Fst translation [66,67]. The group also showed that RNAI SD sequence is sequestered within an intramolecular stem-loop and that translation repression is maintained until interaction with RNAII [68,69]. These studies also demonstrated that, upon pAD1 plasmid loss, RNAII, in complex with RNAI, is degraded by RNases and that Fst is produced from RNAI, resulting in cell death. Indeed, ectopic overexpression of Fst rapidly provokes a chromosome segregation defect evidenced by nucleoid condensation and misplaced septa, resulting in daughter cells with little or without DNA within 15 min after induction in *E. faecalis* (Figure 3) [70]. These cell division abnormalities precede the simultaneous inhibition of DNA replication, transcription, and translation, membrane permeabilization, and thus cell growth nearly 45 min after Fst induction, highlighting that morphological changes are the primary detected effects related to the toxicity of the toxin (Figure 3) [71]. Moreover, Fst overexpression did not lead to the formation of “ghost” cells or leakage of cellular content as for HokB and TisB toxins [9,30,71], confirming that Fst is not a pore-forming toxin. Synthetic Fst has no effect on bacteria, fungal cells, or erythrocytes when added to the extracellular medium [22,71], indicating that Fst is localized within the membrane to facilitate interactions with a specific membrane-bound intracellular target or is intracellularly modified rather than being directed against the membrane itself as observed with antimicrobial peptides. Interestingly, microarray analysis showed that Fst overexpression increases the expression of several membrane transporters 1 h after induction. This expression is detrimental for the bacteria potentially by depleting the pool of intracellular ATP and/or by perturbing the membrane integrity secondary to defects on nucleoid structure, chromosome segregation, and cell division (Figure 3) [72]. CD, NMR, and MD simulation experiments demonstrated that Fst forms an α-helical transmembrane structure between amino acids 4 and 26 with a slight bend at the G15 and a disordered charged C-terminal domain (Table 1). The presence of negatively charged residues (D26 to D30) at the C-terminal domain suggests that Fst binds to the cell membrane with the C-terminus pointing into the cytosol since the outer side of Gram-positive membranes is negatively charged (Figure 1) [22]. The authors hypothesized that Fst anchoring at the membrane facilitates binding with membrane-bound target through the C-terminal domain, which would become structured after binding [22]. However, mutagenesis analysis demonstrated that the Fst C-terminal domain is dispensable for the toxin toxicity, although its deletion seems to reduce toxicity [49]. Conversely, the hydrophobic domain, notably the P11 and the G15 residues present in the conserved motif APXXXGXXX, and the two charged amino acids K2 and D3 at the N-terminus, are essential for the toxicity of Fst [26].

The BsrG type I toxin is a component of the temperature-dependent *bsrG*/SR4 TA system located on the SPβ prophage region of the *Bacillus subtilis* chromosome [73]. The *bsrG* mRNA encodes a 38-amino-acid hydrophobic peptide whose ectopic overexpression causes cell lysis on agar plates. This toxicity is reversed by the bifunctional SR4 RNA antitoxin that interacts with *bsrG* mRNA via its overlapping 3’-end, leading to the RNase III-mediated degradation of the *bsrG*/SR4 duplex and to the *bsrG* mRNA translation inhibition by sequestering the Shine–Dalgarno (SD) sequence [73,74]. To analyze the mechanism of action of BsrG toxin, the *bsrG* gene is integrated into the *aprE* locus of the *B. subtilis* chromosome under the control of an IPTG-inducible promoter. In this condition, the BsrG expression causes a slight growth inhibition of *B. subtilis* 3 h after IPTG induction (Figure 3) [75]. Although the BsrG toxin is associated with the cell membrane, its IPTG-inducible expression induces neither membrane permeabilization and depolarization, nor alteration of membrane fluidity and intracellular ATP level [75]. However, *bsrG* induction provokes morphological abnormalities visualized by 3D-structured illumination microscopy such as reduced cell size, irregular shape, distorted cell division planes, membrane invaginations, and irregular septa [75]. BsrG also induces nucleoid condensation without alterations in chromosome segregation and replication and promotes a global inhibition of transcription and translation as a secondary consequence of its toxicity. After a detailed analysis of the different effects of BsrG overexpression, the authors concluded that BsrG stimulates fatty acids biosynthesis, causing invaginations of the cytoplasmic membrane, leading to the delocalization of the cytoskeletal protein MreB and the accompanied cell wall synthesis machinery. This triggers cell lysis, which is dependent of LytC and LytD autolysins [75]. However, the direct cellular target of BsrG is not the cell wall synthesis machinery and remains to be identified [76]. From a structural point of view, addition of the monomeric superfolder GFP to the C-terminus of BsrG removes the toxicity, suggesting that the cationic and polar C-terminal domain predicted to be located inside the cell (Table 1) is essential for the interaction of BsrG with an intracellular target [75].

The *H. pylori* AapA1 type I toxin is a 30-amino-acid peptide from the *aapA1*/IsoA1 TA system initially discovered by global transcriptome analysis in *H. pylori* 26695 strain [77]. Structural rearrangements of the *aapA1* mRNA leads to a duplex formation with the *cis*-encoded IsoA1 RNA antitoxin [78]. This interaction promotes the translation inhibition of the *aapA1* mRNA and its degradation by RNase III, thus preventing toxin synthesis under normal growth conditions [78]. The ectopic overexpression of AapA1 leads to a growth arrest characterized by a rapid morphological transformation of *H. pylori* from spiral-shaped bacteria to round coccoid cells [78,79]. This transformation occurs as early as 2.5 h after toxin induction that corresponds to the division time of *H. pylori*. AapA1 is exclusively located to the inner membrane [24,79]. The intracellular ATP level and the membrane potential are weakly affected 6 h and 8 h after AapA1 induction, whereas most cells are transformed into viable coccoids (Figure 3) [79]. Experiments using in vitro membrane models, Plasmon waveguide resonance, MD simulations, and Cryo-EM demonstrated that the interaction between AapA1 and cell membrane induces a lipid reorganization and a thinning of the bilayer lipids without severe membrane disruption [24]. The structural analysis performed using NMR shows that the AapA1 toxin folds into three functional domains with an amphipathic α-helical transmembrane segment spanning from S9 to L28, flanked by two positively charged domains that appear to be unfolded (Table 1). The first eight amino acid residues are not required for toxicity, in contrast to the two positively charged residues, K29 and R30, at the C-terminus [24]. The addition of a SPA tag sequence at the C-terminus also suppresses the AapA1 toxicity. The C-terminal part of AapA1 is localized in the cytosol, suggesting that the K29 and R30 residues allow the binding with an intracellular target, in addition to the membrane perturbations, to interfere with cell elongation and division, leading to the formation of viable coccoids [24,79]. Coccoids are induced by oxidative stress, probably through an increase of AapA1 expression consecutive to a drop of IsoA1 RNA level [79]. AapA1 represents an essential effector of the morphological conversion of *H. pylori* from spirals to coccoids observed in human gastric biopsies and compared to persister cells. However, the deletion of the five homologues of AapA toxins in *H. pylori* has no impact on the persister cells formation after oxidative stress exposure [79].

The LdrD toxin is part of the Fst/Ldr family of type I toxins and composed of 35 amino acids. It belongs to the *ldrD*/RdlD TA system located on the *E. coli* chromosome [80]. The RdlD (for Regulator detected in LDR) RNA antitoxin inhibits *ldrD* mRNA translation, but the molecular mechanism of regulation is currently unknown. The ectopic overexpression of LdrD causes a rapid growth inhibition, loss of cell viability, and nucleoid condensation within 2 min upon induction (Figure 3) [80]. Due to the celerity of this morphological modification, the authors indicated that it is unlikely that the nucleoid condensation is linked to the accumulation of LdrD on the chromosome. The authors hypothesized that LdrD interacts with a specific cellular target important for maintaining the integrity of the nucleoid structure and the cell growth [80]. Microarray analysis showed that LdrD induction up-regulates the expression of genes involved in the purine metabolism and decreases the expression of proteins located in the membrane (Figure 3) [80]. It would be interesting to investigate the effect of the LdrD induction on the membrane permeabilization, the membrane depolarization, and the intracellular ATP level as performed for the BsrG and AapA1 type I toxins and for its homologue LdrA (57% identity and 74% homology with LdrD) to decipher the exact mechanism responsible for LdrD toxicity. NMR and CD assays showed that LdrD possesses an α-helical transmembrane domain and binds to phosphocholine micelles without changing their size (Figure 1) [21; PDB id: 5LBJ]. Interestingly, LdrD exhibits two conserved cationic amino acids at its C-terminal domain, which are also found for LdrD, AapA1, Fst, BsrG, and SprG1_31_ and SprG1_44_ toxins. Notably, they are essential for the toxicity of AapA1 toxin also inducing morphological changes as a primary observable effect (Table 1) [24]. Mutagenesis analysis could be performed to identify the amino acids that confer the toxicity of LdrD towards *E. coli*.

## 5. Protein Folding of the *S. aureus sprG1*-Encoded Type I Toxins

We will present here original results concerning the first solution structure of the *Staphylococcus aureus* SprG1-encoded toxin peptide, determined by NMR. The SprG1-encoded type I toxins belong to the *sprG1*/SprF1 TA system located within a pathogenicity island (PI) in *S. aureus* and identified by homology with the *B. subtilis txpA*/RatA type I TA system [8,81]. The *sprG1* mRNA encodes two membrane peptides from a single internal reading frame: a long (44 amino acids, SprG1_44_ also named PepG1_44_) and a short version (31 amino acids, SprG1_31_ also named PepG1_31_). The SprG1_44_ peptide has 13 extra amino acids in the N-terminus compared to SprG1_31_ (Table 1). Ectopic overexpression of both SprG1-encoded peptides causes *S. aureus* cell death accompanied by disruption of membrane integrity within 1 h upon induction (Figure 2) [81]. The dual-function RNA antitoxin SprF1 promotes *sprG1* mRNA degradation and prevents *sprG1* mRNA translation by interacting in *cis* with its overlapping 3’-end [81]. Moreover, thanks to a purine-rich sequence located at its 5′-end, SprF1 also interacts with a subset of polysomes and ribosomes that could promote translation attenuation and persister cell formation [82]. The extracellular addition of chemically-synthesized peptides SprG1_44_ and SprG1_31_ or of membrane extracts prepared from *S. aureus* cells overexpressing SprG1-encoded peptides trigger the lysis of both competing bacteria (Gram-negative and positive bacteria) and human erythrocytes [81]. To better understand the mechanism of action of the SprG1-encoded peptides responsible for membrane permeabilization, we decided to solve their structure by NMR (Section S1, S2) and focused on the short SprG1_31_ peptide (Figure 1). Chemically-synthesized SprG1_31_ peptide is hydrophobic and thus not soluble in water. It dissolves upon addition of deuterated isopropanol. The addition of 50% v/v *d8*-isopropanol avoids the presence of undissolved peptide in the NMR tube and yields acceptable NMR resonances line broadenings. In such conditions, SprG1_31_ peptide adopts a single well-folded conformation based on the TOCSY and NOESY spectra (Figure 4, PDB id: 7NS1). The 3D structure using NMR restraints displays an almost perfect α-helix about 39Å long, ranging from I4 to S28. Most hydrophilic residues are at the N- and C-termini. The three N-terminal MIT residues and the three hydrophilic C-terminal NKK residues are unstructured, with enhanced flexibility for the C-terminus. Albeit not stacked, the two phenylalanine residues (F10 and F13) in the first half of the peptide are about 7 Å from each other. They also share the same strongly hydrophobic side with L14, L17, I18, L20, and V21 (Appendix A). The NMR spectra recorded in the same conditions on the SprG1_44_ peptide disclose large resonances especially in the N-terminal region, indicating that this part of the peptide is not well structured and/or very flexible (Appendix A). Despite a poorer resolution, a good superposition is observed for the amino acids corresponding to the medium part and the C-terminus part of SprG1_31_ peptide (Appendix A). From the similarity of chemical shift and connectivities, we can conclude that the helical structure ranging from L8 to S28 in the SprG1_31_ peptide is retained in SprG1_44_ (i.e., from L21 to S41) (Table 1). In particular, the well-defined part of the α-helix comprising the two phenylalanine residues is conserved. On the opposite, the cross-peaks of the amino acids I15–M20 corresponding to SprG1_31_ I2–M7 are shifted and/or broadened (sometimes broadened out to baseline), and many connections are lost. This indicates that the extra N-terminal amino acids in SprG1_44_ peptide destabilizes the region corresponding to the N-terminus of the SprG1_31_ peptide.

In our experimental conditions, there is no evidence that SprG1_31_ peptide undergoes conformational exchange or multimerizes. However, it is small and dimers, if totally symmetrical, would be undetectable by our NMR experiments. SprG1_31_ and SprG1_44_ peptides are known to accumulate at the *S. aureus* membrane and to be able to lyse competing bacteria and host cells [81]. The last half of the extra N-terminal sequence of SprG1_44_ peptide is mainly hydrophilic and cationic (with a KSLERRR tract), explaining its higher cytolytic effect than SprG1_31_ against human erythrocytes [74]. Although we have no evidence for this, it is possible that the extra N-terminal amino acids of the “long” peptide display a signal sequence that may improve secretion efficiency. SprG1_44_ may also interact with SprG1_31_ and drags its shorter version like a cargo by intermolecular phenylalanine interactions to reach the bacterial or host cell membranes. Their hydrophobicity makes both peptides suitable for insertion in the bacterial and host cell membranes, in line with the experimentally observed lysis. In such a situation, the two phenylalanine of one monomer may each interact with one of the two phenylalanine of another monomer (from SprG1_31_ or SprG1_44_ peptide), thus helping to form a small pore in the membrane [83]. This mechanism, as well as possible transient interactions between both peptide during blood cell travel, are worth further investigations. Although SprG1_31_ and SprG1_44_ peptides could also damage bacterial membranes and erythrocytes through a detergent-like effect, or by interference with membrane-associated functions, we cannot exclude a role of these toxins in morphological changes as nucleoid condensation.

## 6. Concluding Remarks and Future Perspectives

In this review, we present an overview of data on structure and membrane interactions of toxins that belong to type I TA systems. The common features of the membrane-associated type I toxins, a small size less than 60 amino acids, an important hydrophobic character, a putative α-helical transmembrane domain, and the presence of cationic residues generally localized at the C-terminal domain, allow them to directly interact with bacterial membranes that are mostly negatively charged. In most cases, the membrane-associated type I toxins display their toxicity by interfering with membrane integrity through depolarization and/or permeabilization associated with an intracellular ATP drop. Only two type I toxins, HokB and TisB, have been experimentally demonstrated as pore-forming peptides. For HokB, the pore formation is directly linked to persister cells formation, in contrast to TisB. For the other toxins acting on membrane as a primary detected effect, the mode of action remains to be determined by a global biophysical approach. If they do not form pores, the mode of action of these type I toxins may follow the model of membrane disruption via the “carpet” mechanism in which peptides bind to the surface of the membrane and provoke a detergent-like effect as shown for antimicrobial peptides [84]. As exceptions, four type I toxins (Fst, BsrG, AapA1, and LdrD) display their toxicity by inducing morphological changes as primary detected effect prior to membrane disruption. This suggests that these toxins may exert their toxic effect indirectly or in conjunction with other partners. Some peptides have been described as regulating degradation of membrane proteins, stabilizing P-type ATPase transporters, and modulating the activity of two-component systems [85]. The anchoring of these peptides to the membrane could facilitate interactions with targets. When comparing the sequences and structures of membrane-associated type I toxins, it is difficult to see a common mechanism of action. As an example, LdrA and LdrD toxins display a different mechanism of action despite 74% sequence homology (Figure 2 and Figure 3) [80]. Moreover, many studies reporting type I toxin actions were based upon episomal overproduction, which is an inadequate model where the toxin concentration is not controlled and generally much higher than in reality; it could induce off-target effects that would not appear in natural conditions. It can be also noted that, in most cases, the effect of the membrane-associated type I toxins on the bacterial morphology or on the membrane integrity is usually left unexplored, as only one of these effects is studied. Like DinQ or Lpt toxins, both changes can be observed after toxin overproduction and it is tempting to speculate that some toxins may act through both mechanisms to induce growth stasis or cell death. We aimed here at giving a chronology of the effects induced by the overproduction of toxins in order to highlight the primary effects on bacterial cells and to get insights to compare the associated mechanisms of action. Unfortunately, some studies only focused on the consequences of the overexpression at a specific time after induction and did not provide kinetic data. The information about these toxins is thus incomplete. It is also important that some experiments required an extra incubation time after sampling, like for membrane depolarization measurements, which gives a bias of the “time after induction” indicated in Figure 2 and Figure 3. New hypotheses regarding the biological functions of the membrane-associated type I toxins will arise from a better understanding of their mechanisms of action and the identification of their molecular targets. Remarkably, in the case of type V *ghoT*/GhoS TA locus, the GhoT toxin is predicted to be a small membrane protein (57 amino acids) that alters membrane integrity, leading to a drop in PMF and ATP levels, thereby promoting *E. coli* persistence [86].

The membrane-associated type I toxins can be considered as lead compounds for the design of new antimicrobial drugs. Synthetic SprG1_31_, SprG1_44_, and SprA1 have a bactericidal action on *S. aureus, E. coli*, or *P. aeruginosa* when added in the extracellular medium [48,81]. Specific chemical modifications of SprA1 toxin dramatically increase its antibacterial potential and its stability in human serum with limited resistance while considerably reducing its human cell toxicity. This brings the proof of concept that toxins can be transformed into potent antibiotics [87]. Moreover, the peptide ALLRL from the ZorO toxin displays antimicrobial effects against the Gram-positive bacteria *S. aureus* and *B. subtilis* and the fungus *C. albicans* [54]. The DinQ toxin is another promising candidate for the development of anti-cell-envelope antibiotics, notably targeting *E. coli* infections [88]. These strategies could be applied to other type I toxins, notably for the design of new antibiotics, thus providing alternatives to eradicate resistant bacteria as well as persister cells.

## Figures and Tables

**Figure 1 toxins-13-00490-f001:**
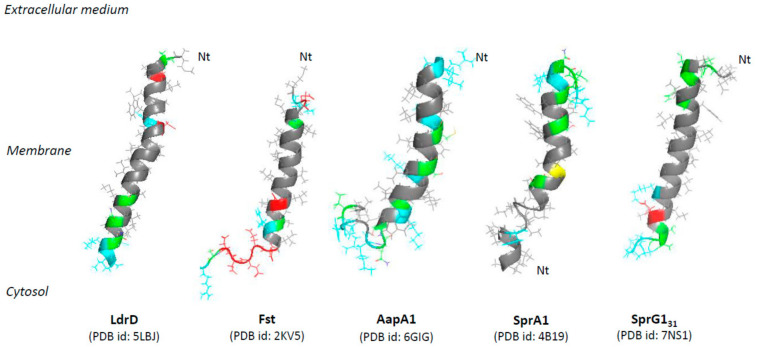
Overview of the structurally determined membrane-associated type I toxins. Hydrophobic amino acids are represented in gray, polar amino acids in green, positively charged amino acids in blue, negatively charged amino acids in red, and cysteines in yellow.

**Figure 2 toxins-13-00490-f002:**
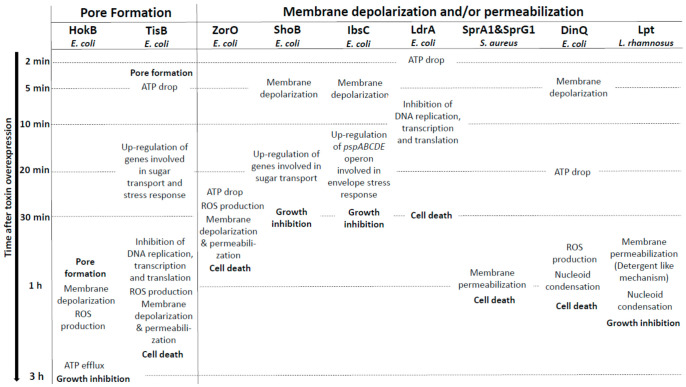
Overview of the membrane-associated type I toxins inducing membrane perturbations as primary detected effect. Only the effects of membrane-associated type I toxins after induction of their overexpression in bacteria have been shown in the figure.

**Figure 3 toxins-13-00490-f003:**
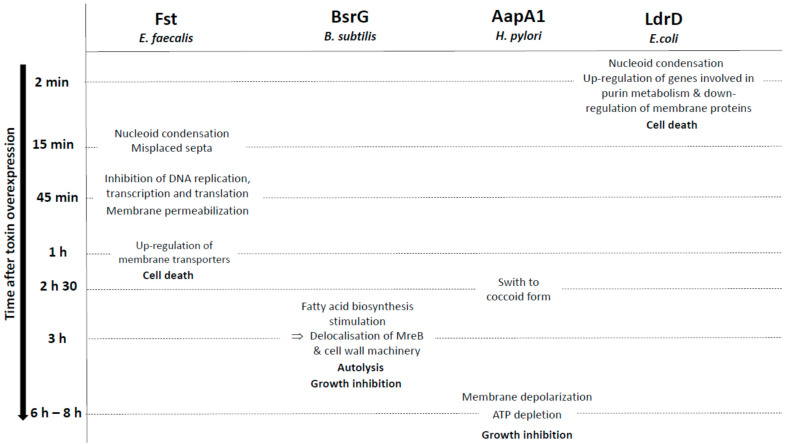
Overview of the membrane-associated type I toxins inducing cell morphology changes as a primary detected effect. Only the effects of membrane-associated type I toxins after induction of their overexpression in bacteria have been shown in the figure.

**Figure 4 toxins-13-00490-f004:**
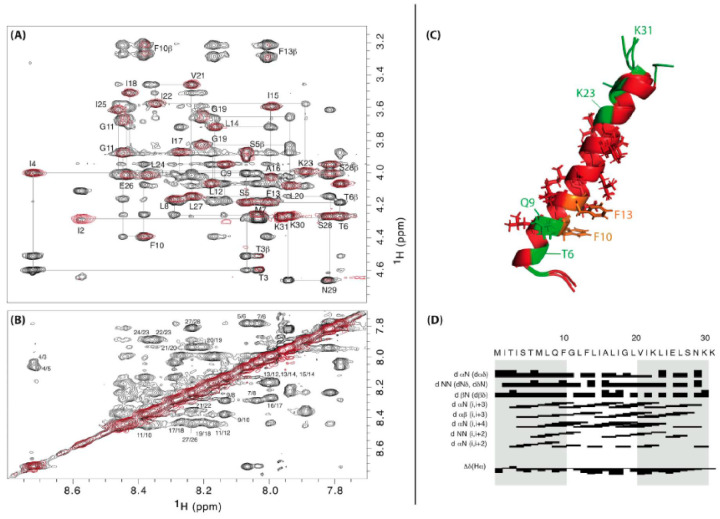
New SprG1_31_ NMR 3D model of structure (PDB 7NS1). (**A**,**B**) Superposition of the 300 ms ^1^H-^1^H NOESY spectrum (black) and the 80ms ^1^H-^1^H TOCSY spectra (red), both recorded at 303K and pH 4.5, showing the NH-Hα (panel **A**) and NH-NH (panel **B**) connectivities. (**C**) Superposition of the five best structures of lower energies and Molprobity scores: hydrophilic amino acids are in green, hydrophobic in red and phenylalanines in orange. (**D**) Distance restraints chart.

**Table 1 toxins-13-00490-t001:** Overview of membrane-associated type I toxins from toxin-antitoxin systems for which insights into their mechanism of action have been published. Predicted or experimentally determined (according to respective PDB file) α-helix are highlighted in orange and β-sheet in yellow. When the structure has not been experimentally determined, α-helix have been predicted with Jpred4 (http://www.compbio.dundee.ac.uk/jpred/index.html, accessed on 5 April 2021) [27]. Transmembrane domains are delimited by lipid representation surrounding each sequence and have been predicted with TMPRED (https://embnet.vital-it.ch/software/TMPRED_form.html, accessed on 5 April 2021). Boxes colored in blue correspond to the toxins inducing morphological changes as a primary detected effect, the green one for toxins inducing membrane perturbations as a primary detected effect and the grey one is for toxins with dual effects. Polar amino acids are shown in green, negatively charged amino acids in red and positively charged amino acids in blue. Cysteins are shown in orange. The charge and the hydrophobicity index (based on Kyte-Doolittle scale) have been calculated thanks to the R package « Peptides » [28,29]. Unexpected results like low hydrophobicity or global negative charge have been written in red.

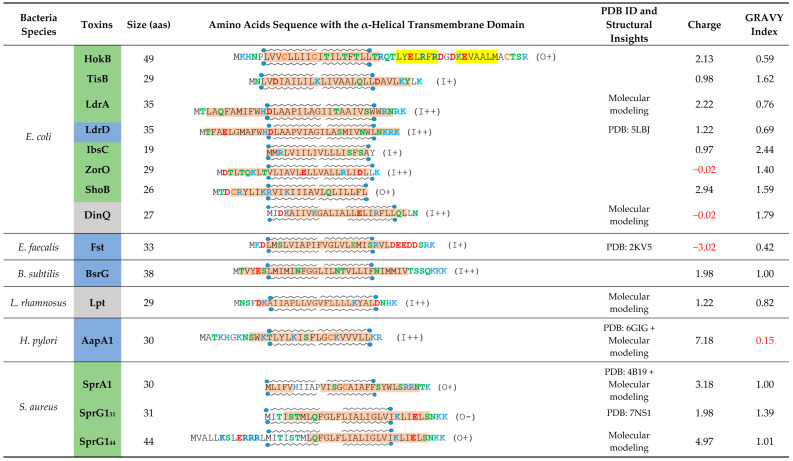

## Data Availability

Not applicable.

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
