# Peer review of "Bacterial Type I Toxins: Folding and Membrane Interactions"

_toxins, 2021, doi:10.3390/toxins13070490_

Round 1
Reviewer 1 Report
Reviewer Comments to Author:
Title: Bacterial Type I Toxins: Folding and Membrane Interactions
Journal: Toxins
The review entitled ” Bacterial Type I Toxins: Folding and Membrane Interactions” summarize the knowledge regarding the bacterial type I toxin-antitoxin systems; especially, membrane-associated type I toxins inducing: pore formation, membrane depolarization and/or permeabilization, cell morphology changes as primary toxic effect and protein folding of the S. aureus sprG1-encoded type I toxins. The Review describe and present in details the respective points, however, some aspects should be clarified, specified and discussed. During the revision some view being difficult to understand. Therefore, please follow the comments and remakes bellow in order to allowed the reader to follow easy the idea:
- The abstract and section 4, underlines the information regarding the enumerated mechanism and also specify the S. aureus strain while section 2, 3 the mechanism about S.aureus, E. Coli, E, faecalis, B.subtilis, H.pylori etc. Moreover, the introduction section specified the investigation of the expression in S. aureus. This aspect makes the reader a little bit confused. The present Review is focused on the type I toxins characteristic only for S. aureus or different bacterial strain? In this context please supplement the introduction section with the information regarding the following aspects: why such bacteria strains (as present the table 2 and 3)? Why table 2 presents information regarding almost E coli while table 3, divers one? Is this aspect connected with the structure, composition (means gram – and gram +) of the respective bacterial strains and action mechanism presented in this study? Please improve the introduction section in this aspect.
- Another aspect that should be explained is the section presented separately regarding the Protein folding of the S. aureus sprG1-encoded type I toxins. The respective section present the own results? Have been this results published before? If yes, please discuss the results as a date obtained by your group that was published in 2019, for ex. Should not be presented as: “In our experimental data….”. The Review should include the date revised from literature data. Please treat your data as a results presented by your group and published by….in…., and specify the name and year. Moreover, the present mechanism is characteristic only for S. aureus or another gram + strains? Please provide to the Review the missed explanation in order to avoid any misunderstanding.
- Therefore, according to the comment in point 2, is appear the question regarding the Fig 4. Did the author prepare this figure especially for the present research based on unpublished data or the figure have been taken from the references?
- After following the comments above, please correlate the abstract, introduction sections in context of the main goal of the Review and the membrane-associated type I toxins mechanism inducing: pore formation, membrane depolarization and/or permeabilization, cell morphology changes regarding the reviewed bacterial strains as it was well done in the conclusions section.

Reviewer 2 Report
This manuscript is a timely review of bacterial type I genetic modules that are two-component systems that encode a toxic protein and an unstable RNA antitoxin that controls toxin translation during cell growth. The type I toxins are small, hydrophobic, alpha-helical peptides with a single transmembrane domain that perturbs membrane function via depolarization and/or permeabilization which can lead to reduction in intracellular ATP levels, growth adaptation and even persister cell formation. This review focusses on the folding and membrane interactions of the type I toxins and charts their toxic effects on bacterial cells. The NMR structure of the sprG1-encoded membrane peptides expressed in S. aureus are described along with some putatitive membrane interactions leading to cell lysis.
General Comments:
- I don't agree with the idea to classify a subset of type I toxins by their ability to induce cell morphology changes as a primary toxic effect, since the morphological change is not the cause of the toxic effect, but rather more of an observed effect as a result of other mechanisms (according to Figure 3). I suggest a change in wording to reflect this.
- The review is light in the area of the influence of antitoxin regulation of toxin expression in these systems. I would suggest providing some information on this topic.
Specific Comments:
- Table 1: the color scheme used to highlight the different categories or toxic effects should exhibit better contrast as it is difficult to differentiate between light blue/green/grey colors.
- Commas are used in some instances instead of decimal points.
- Section 3.3 should be Section 4 and then every subsequent section number should be increased accordingly. This is based on the sentence, "We can distinguish type I toxins whose overexpression induces..." from page 3, before Fig. 1. If distinguishing two subsets of toxins, one should not fall under the same section as the other.
- L10 in the abstract should read "induces" not induce.
- L51: "Most type I antitoxins" rather than "The majority of type I antitoxins".
- L80: "like" should replace "similar to"
- L141: "later" rather than "later on"
- L214: (i.e., Ist-R1...)
- L277: Replace "in order to" with "to"
- Line287: "evidence" instead of "evidences"
- Line324: "Gram-negative"
- Line360: "These results" should replace "This results"
- Line481: Replace "the majority of cells" with "most cells"
- Line542: "peptides trigger" rather than "peptides triggers"
- Line560: Replace "fairly good" with "good"
- Line 564: (i.e., from L21...)
Round 2
Reviewer 1 Report
Thank you very much for such clear and concis explanation. I appreciate the authors effort to improve the present Review. The answers are point by point clearly presented and the manuscript supplimented with the required explanation.